# Identification of Urban and Wildlife Terrestrial Corridor Intersections for Planning of Wildlife-Vehicle Collision Mitigation Measures

**Andrius Kučas** [1,*] , **Linas Balčiauskas** [2] **and Carlo Lavalle** [1]

1   European Commission, Joint Research Centre, Via Fermi 2749, 21027 Ispra, Italy; carlo.lavalle@ec.europa.eu
2   Nature Research Centre, Akademijos Str. 2, 08412 Vilnius, Lithuania; linas.balciauskas@gamtc.lt
*   Correspondence: andrius.kucas@ec.europa.eu; Tel.: +39-0332-78-9445

**Abstract:** Roadkill and other impacts of roads on wildlife create pressures on society and the environment, requiring the implementation of mitigation measures in response. Due to various natural and anthropogenic causes, the locations of wildlife–vehicle collisions are not stable in time and space. The identification of urban and wildlife corridor intersections can help anchor collision locations along high-risk road sections. Urban and wildlife corridors and their intersections were identified in a case study of Lithuania using a landscape connectivity identification method based on circuit theory. A strong relationship was found between the numbers of urban–wildlife corridor intersections and the numbers of wildlife–vehicle collisions. Short road sections were characterised by the number of urban–wildlife corridor intersections, mammal–vehicle collisions, and the presence of fencing. Multi-criteria analyses identified the road sections where wildlife fencing is, simultaneously, the longest, and the number of mammal–vehicle collisions and the number of urban–wildlife corridor intersections are highest. The results show that identifying wildlife and urban corridor intersections can reinforce locations for permanent roadkill mitigation measures. The identification of crossing structure type and location within shortlisted road sections and evaluation of their efficiencies remain the challenges for field research.

**Keywords:** roadkill analysis; resistance; conductance; suitability; permeability; ranking; road impact mitigation

## 1. Introduction

The ecological effect of road traffic on wildlife populations is a growing concern in many countries [1–10]. The rapid growth of urban areas is linked with the worldwide expansion of linear infrastructure networks and poses a global threat to biodiversity and ecosystem services. The ecological and habitat fragmentation caused by road infrastructures can be considered one of the main drivers of wildlife–vehicle collision occurrence [11–14]. Wildlife–vehicle collisions are a cause of serious concern for road planners and biologists in terms of traffic safety, species conservation, and animal welfare [15]. The rate of successful wildlife crossings might decrease significantly [16], despite the wildland–urban interface expansion [17,18], as road networks densify [19] and current roads are upgraded to accommodate higher traffic volumes.

Wildlife–vehicle collisions occur where human and wildlife pathways intersect. Collision hotspot identification tools and techniques can help to identify collision clusters [20,21]. However, collision hotspots are not permanent; their times and locations shift due to changes in wildlife population sizes, wildlife behaviour, landscape fragmentation, and even due to installed collision mitigation measures [22,23]. Installed collision mitigation measures might also distract wildlife and change animal behaviour and the direction in their pathways. New pathways can appear and increase collision rates on more distant

roads that have no mitigation measures installed [20,23]. In order to anchor collision locations or collision hotspots to multispecies collision mitigation measures, it is important to identify high-risk road sections [11,23–26].

There are many types of measures to potentially reduce wildlife mortality on roads [27]. The mitigation measures include overpasses and underpasses [27–32], passages, bridges, culverts, tunnels, poles [27], fencing, gates, crash barriers, wildlife jump-outs [27,29,33] chemical repellents [34,35], reflectors [27,36,37], auditory deterrents [37], and ultrasound [38]. Other types of measures to reduce traffic volume and speed include dynamic driver warning systems (animal detection systems), driver training programs, and wildlife warning signs [23,39]. They have different efficiencies and can be generic or species-specific and used alone or in combination with other mitigation measures [11,39,40]. Thus, various wildlife overpasses (landscape bridges, multi-use overpasses, canopy crossings) and underpasses (viaducts, large mammal underpasses, underpasses with water flow, modified culverts, tunnels, and multi-use underpasses) facilitate connections between habitats and wildlife populations, while measures such as collision detection systems, speed reduction signs, and reflectors are focused on improving driver safety [41].

Landscape-related and human-related variables are subject to change over time, leading to changes in species presence and movement patterns [42,43]. The identification of spatial intersections of wildlife and human corridors can help reinforce collision locations for permanent roadkill mitigation measures, especially in well-connected natural areas where animal mortality rates are higher [44]. Different methods from the fields of urban and territorial planning, landscape ecology, and urban ecology can be used for the identification and evaluation of urban [45–49], ecological [20,50–53], and urban–ecological [54–59] corridors. For the identification of wildlife and urban corridors, we chose to use a circuit-theory-based corridor identification paradigm that allowed for us to model landscape connectivity for different domains.

The aim of this research was to (i) model urban and wildlife corridors in Lithuania; (ii) identify urban–wildlife corridor intersections; (iii) characterise road sections with the number of urban–wildlife corridor intersections, the number of mammal–vehicle collisions, and total length of fences; and (iv) identify high-risk road sections for the deployment of permanent multispecies collision mitigation measures.

## 2. Materials and Methods

### 2.1. Study Area, Road, and Fencing Data

Our study area covered the entire territory of Lithuania (Figure 1), which is characterised mostly as a plain. It covered a surface area of 65,286 square kilometres. In 2012, 33% of the surface was occupied by arable land and permanent crops, 27% by semi-natural vegetation, 33% by forested land, 3% by artificial areas, and 4% by water bodies and other land. The land-cover change (0.48% change rate per year) in the country is slowing, mainly because of the rapid decrease in the intensity of forest conversions [60].

In 2017, there were 21,244 km of the state-owned roads (excluding roads in the cities): a total of 1751 km of the main roads, 4925 km of national roads, and 14,568 km of regional roads [61]. In this study, we analysed the main roads, where over the period from 2002 to 2017, the annual average daily traffic increased from 5600 to 11,000 vehicles per day.

Wildlife fences are the most common wildlife–vehicle collision mitigation measure in Lithuania, with a total length of 803.5 km, or 3.78% of all roads fenced in 2017 (Figure 1). There were 1088 segments of wildlife fences, 680 of which (743.8 km) were implemented on the main roads [23]. In addition to wildlife fences, other collision mitigation measures (underground passages, tunnels, gates, and jump-outs) were implemented. So far, integrated fencing and overpass structures have not been implemented in the country.

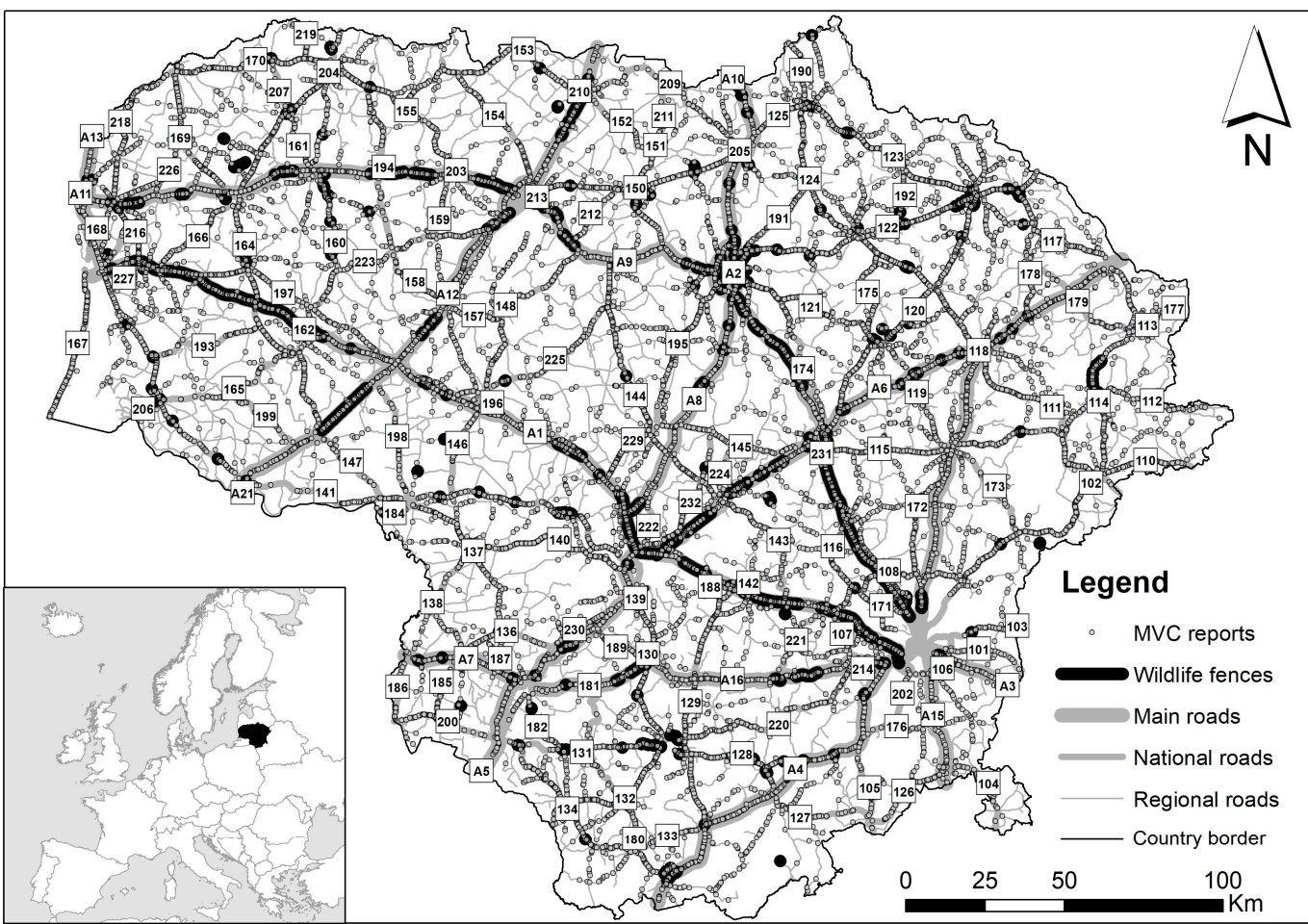

**Figure 1.** Study area. The main roads (lower categories like the national and regional roads that are shown as the reference), fenced road sections, and the locations of mammal–vehicle collisions in the years from 2002 to 2020. Main road numbers are placed within the white-coloured rectangle.

## 2.2. Wildlife-Vehicle Collision Data

In this study, we used data from the Lithuanian Police Traffic Supervision Service [62] and the Nature Research Centre. According to the official data from the Lithuanian Police Traffic Supervision Service and the professional observations of the Nature Research Centre, a total of 27,270 mammal–vehicle collisions involving 32 mammal species were recorded over the period from 2002 to 2020 in Lithuania (Table 1), out of which 3679 (13.49%) were within fenced road sections and 100 metres from both ends of the fence.

## 2.3. Preparation of Urban and Wildlife Resistance Grids

We created two separate 20-metre spatial resolution grids covering the whole country of Lithuania. One grid was used for the characterising of urban sprawl resistances and later for identifying urban corridors (urban least-cost pathways), while another grid was created for characterising wildlife (mammal focus) resistances and identifying wildlife corridors (wildlife least-cost pathways). Every cell in both grids was characterised with CORINE land cover 1:50,000 (reference year 2018) (CLC) [63] land-cover/land-use classes. Later, every cell was characterised with land-cover suitability and permeability scores. Every cell received habitat suitability scores ranging from 1 to 3, where: 1—the core habitat, 2—the edge habitat, and 3—not suitable for habitat. Every grid cell also received land-cover permeability scores that also ranged from 1 to 4, where: 1—small resistance, 2—resistance is smaller than permeability, 3—resistance is higher than permeability, and 4—land cover/land use is not permeable. Grid cells in the urban resistance grid were characterised with urban suitability

and permeability scores, while cells in the wildlife resistance grid were characterised with wildlife suitability and permeability scores (Table 2). Each cell in both resistance grids was attributed with a score reflecting the energetic "cost"—difficulty to inhabit or move across that cell [51]. This scoring mechanism allowed for the creation of urban and wildlife habitat intermediate resistance grids later used for the identification of least-cost paths (corridor links or corridors).

**Table 1.** The number of mammal–vehicle collisions in Lithuania, registered in 2002–2020. Species included in the national Red data list are marked with an asterisk.

| Species [1] | Number of MVC |
|---|---|
| Roe deer (*Capreolus capreolus*) | 17,825 |
| Moose (*Alces alces*) | 1961 |
| Wild boar (*Sus scrofa*) | 1702 |
| Raccoon dog (*Nyctereutes procyonoides*) | 1519 |
| Eastern European hedgehog (*Erinaceus concolor*) | 1093 |
| Red fox (*Vulpes vulpes*) | 1031 |
| European hare (*Lepus europaeus*) | 609 |
| Red deer (*Cervus elaphus*) | 491 |
| Marten (*Martes* sp.) | 408 |
| European polecat (*Mustela putorius*) | 174 |
| Badger (*Meles meles*) | 156 |
| Pine marten (*Martes martes*) | 59 |
| Beaver (*Castor fiber*) | 38 |
| Fallow deer (*Dama dama*) | 35 |
| American mink (*Neovison vison*) | 26 |
| Stone marten (*Martes foina*) | 24 |
| Eurasian otter (*Lutra lutra*) * | 19 |
| Gray wolf (*Canis lupus*) | 11 |
| European bison (*Bison bonasus*) * | 7 |
| Mountain hare (*Lepus timidus*) * | 1 |
| Lynx (*Lynx lynx*) * | 1 |

[1] accidentally road-killed species: red squirrel (*Sciurus vulgaris*), European mole (*Talpa europaea*), Norway rat (*Rattus norvegicus*), bank vole (*Clethrionomys glareolus*), common shrew (*Sorex araneus*), muskrat (*Ondatra zibethicus*), yellow-necked mouse (*Apodemus flavicollis*), stoat (*Mustela erminea*) *, least weasel (*Mustela nivalis*), water shrew (*Neomys fodiens*), and black rat (*Rattus rattus*).

Additionally, urban and wildlife resistance grids were characterised with land-cover permeability scores using a large spatial scale: State Cadastre of Protected Areas of the Republic of Lithuania 1:10,000 (STK) [64], a spatial dataset of (geo) reference base cadastre 1:10,000 (GRPK) [65], administrative boundaries and addresses (RC) 1:500 [66] and annual average daily traffic 1:1000 (LAKD) [67] feature datasets (Table 3). After the spatial union of all feature datasets, grid cells in the urban resistance grid received urban permeability scores, while cells in the wildlife resistance grid received wildlife resistance scores (Table 3). Every grid cell received land-cover permeability scores that range from −1 to 4, where: −1—no barriers (or conductance), 1—small resistance, 2—resistance is smaller than permeability, 3—resistance is higher than permeability, and 4—land cover/land use is not permeable.

Then, all scores in every urban grid cell were summed up, and the final urban resistance grid was created (Figure 2A). The same score calculations were performed for the preparation of the wildlife resistance grid (Figure 2B).

**Table 2.** List of land-cover/land-use suitability and the permeability scoring code list.

| Corridor Type | Urban | | Wildlife | |
|---|---|---|---|---|
| **Land-Cover Classes** | **Suitability** | **Permeability** | **Suitability** | **Permeability** |
| Discontinuous urban fabric | 1 | 1 | 3 | 3 |
| Continuous urban fabric | 1 | 1 | 3 | 4 |
| Industrial or commercial units | 1 | 1 | 3 | 4 |
| Road and rail networks and associated land | 1 | 1 | 3 | 4 |
| Port areas | 1 | 1 | 3 | 4 |
| Airports | 1 | 1 | 3 | 4 |
| Dump sites | 1 | 2 | 3 | 3 |
| Construction sites | 1 | 2 | 3 | 3 |
| Green urban areas | 1 | 2 | 3 | 3 |
| Sport and leisure facilities | 1 | 2 | 3 | 3 |
| Non-irrigated arable land | 2 | 2 | 2 | 2 |
| Permanently irrigated land | 2 | 2 | 2 | 2 |
| Fruit trees and berry plantations | 2 | 2 | 2 | 2 |
| Pastures | 2 | 2 | 2 | 1 |
| Annual crops associated with permanent crops | 2 | 2 | 2 | 1 |
| Complex cultivation patterns | 2 | 2 | 2 | 1 |
| Land principally occupied by agriculture, with significant areas of natural vegetation | 2 | 2 | 2 | 1 |
| Agro-forestry areas | 2 | 2 | 2 | 1 |
| Broad-leaved forest | 3 | 3 | 1 | 1 |
| Coniferous forest | 3 | 3 | 1 | 1 |
| Mixed forest | 3 | 3 | 1 | 1 |
| Natural grasslands | 3 | 3 | 1 | 1 |
| Transitional woodland shrub | 3 | 3 | 1 | 1 |
| Beaches, dunes, sands | 3 | 3 | 1 | 2 |
| Sparsely vegetated areas | 3 | 3 | 1 | 2 |
| Burnt areas | 3 | 3 | 1 | 2 |
| Inland marshes | 3 | 4 | 2 | 3 |
| Peat bogs | 3 | 4 | 2 | 3 |
| Water courses | 3 | 4 | 3 | 3 |
| Water bodies | 3 | 4 | 3 | 3 |

**Table 3.** List of land-cover/land-use permeability scores for different feature datasets.

| Source | Feature Datasets | Permeability | |
|---|---|---|---|
| | | **Urban** | **Wildlife** |
| RC | Building footprints | −1 | 4 |
| LAKD | The main roads/highways | −1 | 4 |
| GRPK | Railroads | −1 | 3 |
| LAKD | National roads | −1 | 3 |
| LAKD | Regional roads | −1 | 2 |
| LAKD | 70, 50, and 20 metre buffer areas from main, national, and regional roads, respectively | 1 | 1 |
| GRPK | Urban (build-up) areas | 1 | 3 |
| LAKD | Wildlife underpasses | 2 | −1 |
| LAKD | Fences against animals | 2 | 3 |
| STK | Protected sites and NATURA2000 areas | 3 | −1 |
| GRPK | Inland waters, lakes and rivers, marshes | 4 | 3 |

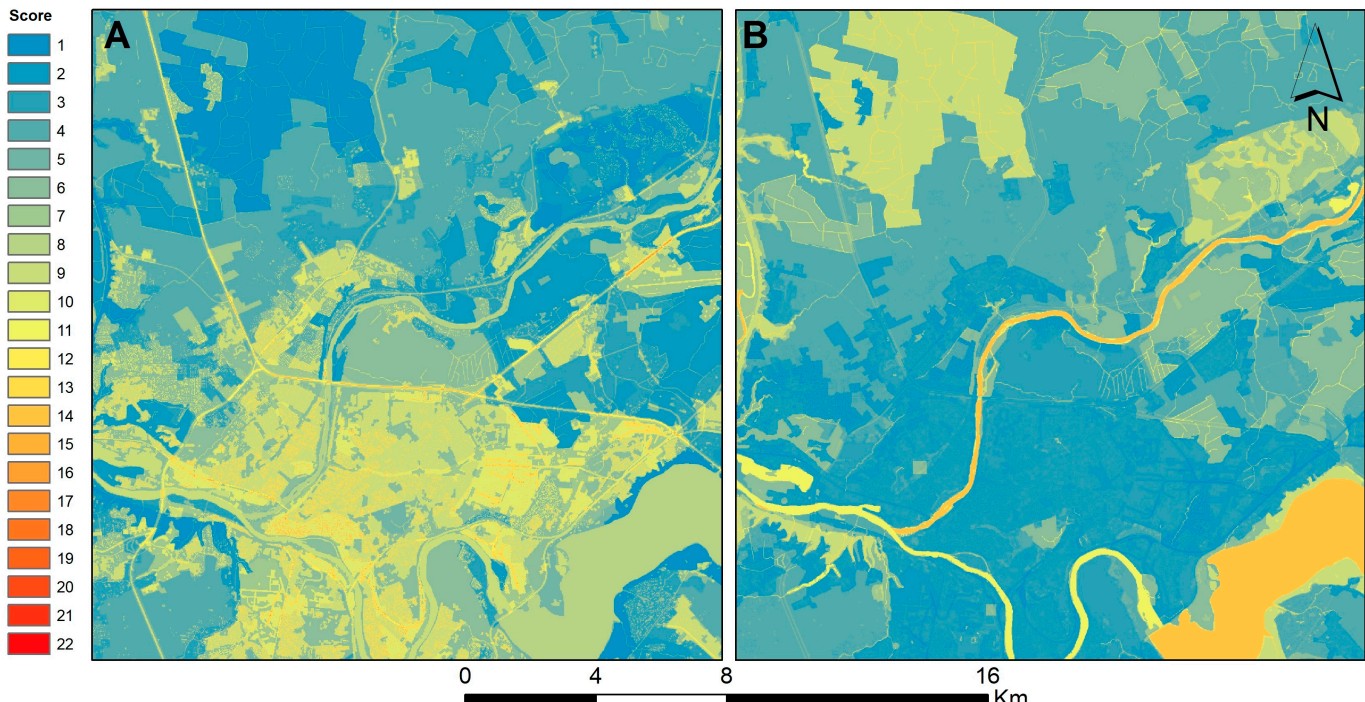

**Figure 2.** Example of wildlife (**A**) and urban (**B**) resistance grids (a higher total score indicates higher resistance) in the northern agglomeration of the Kaunas city.

　　Every grid cell within the wildlife resistance grid was calculated using the following method: the cell that falls within the discontinuous urban fabric land-cover class and is characterised by the presence of a fenced regional road with moderate traffic intensity and an underpass installed receives the final score of 11 (Table 4). For the urban resistance cell score, it is 6. From a land-cover resistance point of view, this cell is less resistant to urban sprawl and more resistant to wildlife. Conductance scores of −1 would lower the final resistance score in both grids.

**Table 4.** Example of grid cell total resistance score calculation for wildlife and urban resistance grids.

| Grid Cell Permeability Characteristics | Scores Assigned | |
|---|---|---|
| | **Wildlife Resistance Grid** | **Urban Resistance Grid** |
| Discontinuous urban fabric * | 3 | 1 |
| Discontinuous urban fabric | 3 | 1 |
| Regional roads | 2 | −1 |
| Annual average daily traffic | 1 | 1 |
| Wildlife underpasses | −1 | 2 |
| Fences against animals | 3 | 2 |
| Final score (sum of scores) | 11 | 6 |

* Land-cover suitability.

　　Following the Linkage mapper tool recommendations [51], wildlife (Figure 2A) and urban (Figure 2B) resistance grids were normalised. Both urban and wildlife resistance grid score values (that ranged from 1 (no resistance) to 22 (complete resistance)) were reclassified to a new range (from 0 (no resistance) to 100 (complete resistance)) and then used as an input for the preparation of cost-weighted distance grids.

　　For the identification of corridors using the circuit-theory-based method, it was necessary to use core areas (nodes). Thus, larger-than-average-sized urban (build-up areas, GRPK in Table 3) areas were used as urban core areas or nodes, while larger-than-average-sized protected sites and NATURA 2000 areas were selected and used as wildlife core areas (STK

in Table 3, Figure 3). We chose larger-than-average areas because we wanted to identify only the core corridors that connect major urban areas (in the case of the urban corridors) and major protected areas (in the case of the wildlife corridors), because multispecies fencing and crossing structures are designed to create a passage for the majority of (but not all) mammal species on the main roads, which are characterised by higher speed and traffic intensity.

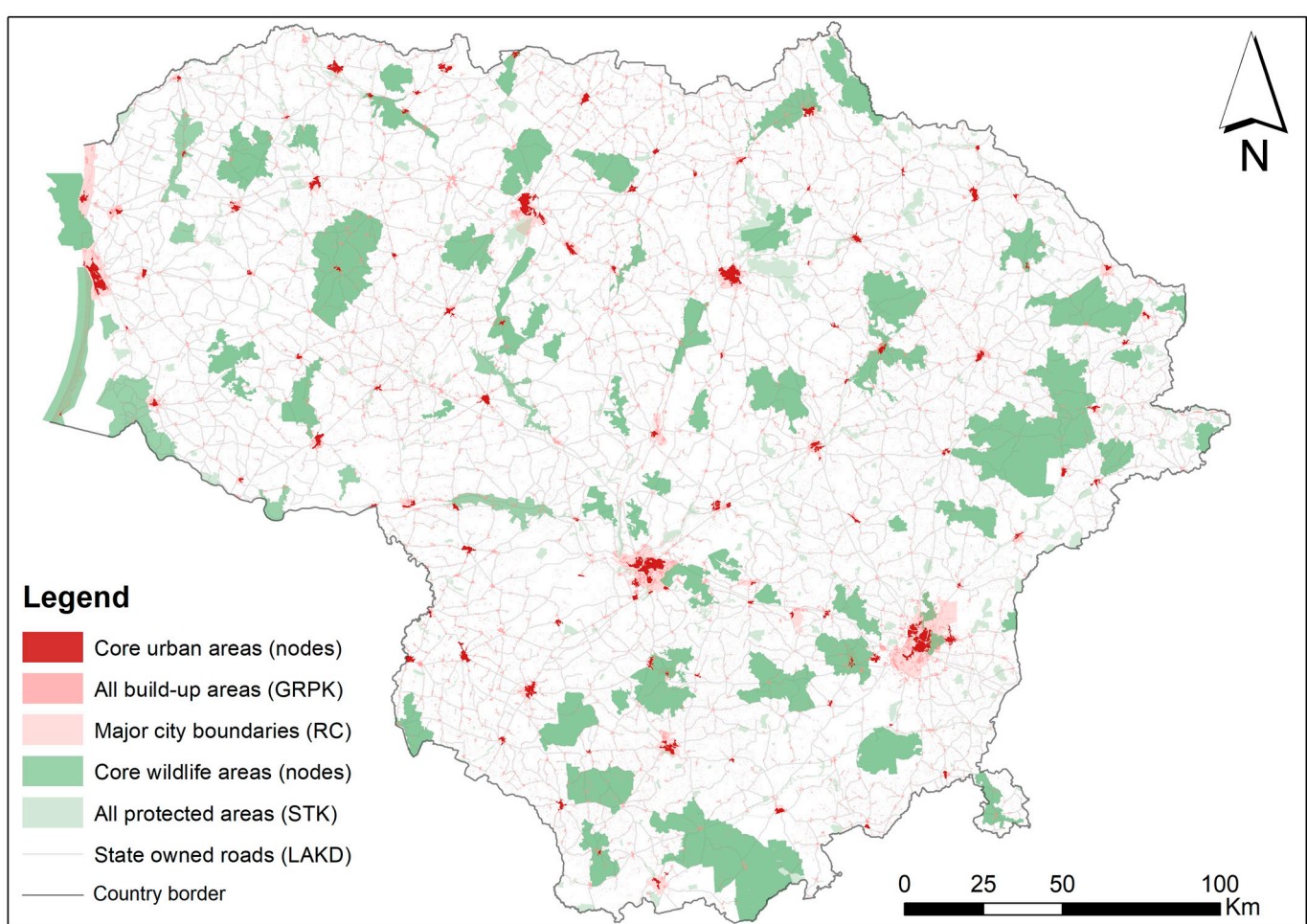

**Figure 3.** Urban and wildlife core areas used as nodes for urban and wildlife corridor identification. Build-up, city boundary, protected areas and state-owned roads are shown as reference (Table 3).

### 2.4. Identification of Terrestrial Urban and Wildlife Corridors

For the identification of urban and wildlife corridors, we used a least-cost path modelling method that combines circuit theory (in physics) with random walk theory [51,52,68]. This method allowed for us to identify multiple diffusion paths, indicate corridor redundancy, and determine the relative importance of habitat patches (nodes) and corridors (links) by the strength of currents between source and ground nodes (Figure 3). Cost-weighted distance grids were created using the circuit theory current (estimates of the net movement probability of a random wanderer through a given cell [51]). The least-cost distance grids (Figure 4A,C) were created by applying the minimum values of the path from cost-weighted distance grids. Finally, by using least-cost distance grids, the least-cost path vector lines (corridor links, Figure 4) were identified and mapped [51,52,68].

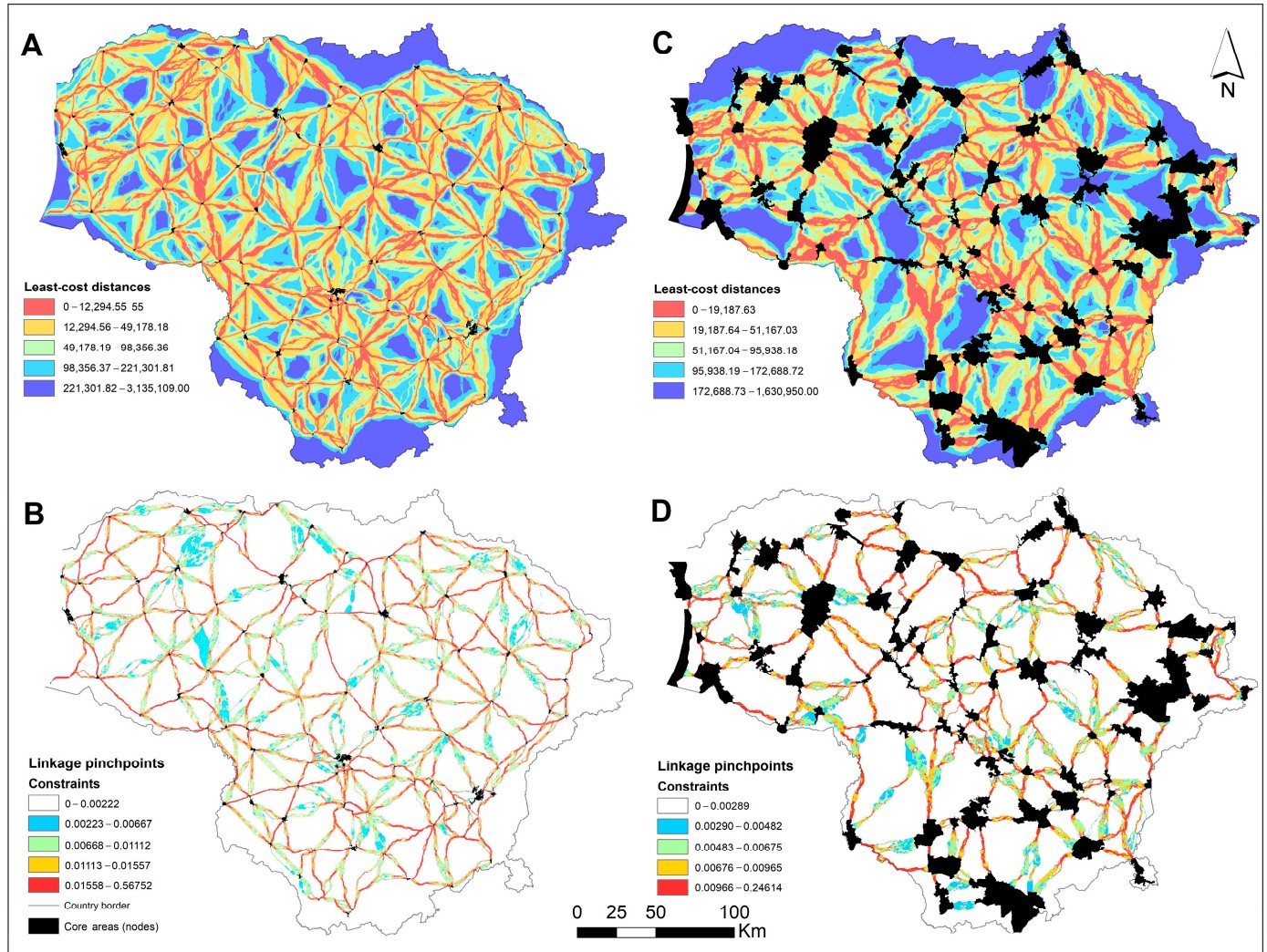

**Figure 4.** Urban corridor least-cost distance (**A**), urban pinch point (**B**), and wildlife corridor least-cost distances (**C**) and wildlife pinch-point (**D**) quantile maps. Urban (**A**,**B**) and wildlife (**C**,**D**) core areas are coloured in black.

We identified wildlife corridors using a wildlife resistance grid (Figure 2A) that connects wildlife core areas/nodes (Figure 3). Urban corridors are identified using an urban resistance grid (Figure 2B) that connects core urban core areas/nodes (Figure 3).

Wildlife and urban pinch points (bottlenecks) also have been identified (Figure 4B,D). Wildlife pinch points with high current values represent bottlenecks in the corridor and indicate a high risk of vulnerability for wildlife pathways. Urban pinch points indicate core urban agglomeration bottlenecks. Urban pinch points with high current values represent bottlenecks in the corridor and indicate locations that are fragile but critical to upkeep the network of urban corridors connected.

Circuitscape (pairwise mode) backed the Linkage Mapper toolbox [51] for ESRI inc. ArcGIS Desktop was used for the identification of intermediate urban and wildlife cost-weighted distances, least-cost distances, and least-cost paths. Pinchpoint Mapper (another tool within the Linkage Mapper toolbox that is based on a current flow model) was used to identify pinch points that show the links that limit movement (wildlife corridors) or urban sprawl (urban corridors) due to unfavourable land-cover/land-use resistances.

### 2.5. Identification, Characterisation, and Ranking of Road Sections

The average spacing between core wildlife bridges (overpasses) may range from around 12.0 km [69] to 22.5 km [70,71]. Therefore, we split all main road network (Figure 1) into smaller road sections and characterised them with (i) the number of mammal–vehicle collisions (Figure 1), (ii) the total length of wildlife fences (Figure 1) and (iii) the total number of urban–wildlife corridor (least-cost path line) intersections (that are found within a 25 km buffer from the road section).

The ranking of road sections was performed using the technique for order of preference by similarity to the ideal solution (TOPSIS) method [72] implemented in SortViz for ArcGIS Desktop [73,74]. Using this method, we chose road sections as alternative options for the ranking and the number of mammal–vehicle collisions, total fence length and the number or urban–wildlife corridor intersections as criteria. Later, we assigned utility functions (maximisation) and weights to each criterion. The criteria weights: (i) the number of mammal–vehicle collisions (0.268), (ii) total fence length (0.448), and (iii) the number of urban–wildlife corridor intersections (0.284) have been defined and applied using automatic criteria weight identification functionality within SortViz for ArcGIS Desktop. This machine-based criteria weighting [73] allowed for us to avoid subjective judgements about the relevance of each criteria in the overall ranking framework, favouring an objective evaluation of road sections based on input data scattering.

The main road sections that simultaneously had (i) the highest numbers of mammal–vehicle collisions, (ii) the longest fences, and (iii) the largest number of urban–wildlife corridor intersections received the highest-ranking scores (in the range from 0–worst to 1–best). The main road sections with the highest TOPSIS score are the most economical and preferable sections where fencing and crossing structures (overpasses for mammal species) should be installed first.

Standard Microsoft Inc. Excel [75] and ESRI Inc. ArcGIS Desktop, ArcMap [76] software packages were used for the road section identification, characterisation, and mapping.

## 3. Results

### 3.1. Urban and Wildlife Resistance Grids and Nodes

The highest wildlife resistance scores within the wildlife resistance grid (Figure 2A) were assigned to the road-land-cover classes that are adjacent to or within urban areas, whereas other land-cover classes such as forests and shrubs received the lowest resistance scores. The highest urban agglomeration resistance scores (Figure 2B) were assigned to water bodies and natural areas, while the artificial surfaces such as build-up areas and linear infrastructure (e.g., roads, railways) received the lowest resistance scores.

For the identification of urban corridors, we used 109 core urban areas (nodes), with an average area of 4.04 km$^2$ and a maximum distance of 28.5 km between them. For the identification of wildlife corridors, we used 65 core wildlife areas (nodes) with an average area of 143.64 km$^2$ and a maximum distance of 33.1 km between them (Figure 3). Finally, using core areas and resistance grids, both urban (Figure 4A) and wildlife (Figure 4C) least-cost distance-based corridor grids were created using the Circuitscape Linkage Mapper tool (set to run in pairwise mode).

### 3.2. Urban and Wildlife Corridors, Pinch Points and Intersections

Urban least-cost distances (Figure 4A) and pinch points (Figure 4B), as well as wildlife least-cost distances (Figure 4C) and pinch points (Figure 4D) for the entire study area, were used to identify least-cost path intersections. Pinch-point areas classified as highly constrained (Figure 4B,D) indicate locations that are important for sustaining respective network connectivity.

The result (Figure 4) shows the relative value of each grid cell in providing connectivity between core areas, allowing for us to identify which routes encounter more or fewer features that facilitate or impede movement between core areas. As animals (wildlife corridors) and humans (urban corridors) move away from specific core areas, cost-weighted

distance analyses produce grids of total movement resistance accumulated. The Linkage Mapper tool then identifies adjacent (neighbouring) core areas and creates grids of least-cost distance (in metres) corridors between them (Figure 4). The pinch points represent areas where movement would be funnelled and thus may be particularly important to keep intact. Even a small loss of area in these pinch points (for instance, fencing the road without the installation of crossing structures) would disproportionately compromise connectivity [51,77] and negatively impact populations [78].

Both urban and wildlife least-cost path grids (respectively, Figure 4A and 4C) are converted to vector lines (corridor links) by simplifying a line and removing small fluctuations or extraneous bends. Finally, urban and wildlife least-cost paths intersected (Figure 5). In total, we found 354 intersections of urban and wildlife least-cost paths that indicate potentially stable locations where urban and wildlife least-cost paths may encounter each other (Figure 5).

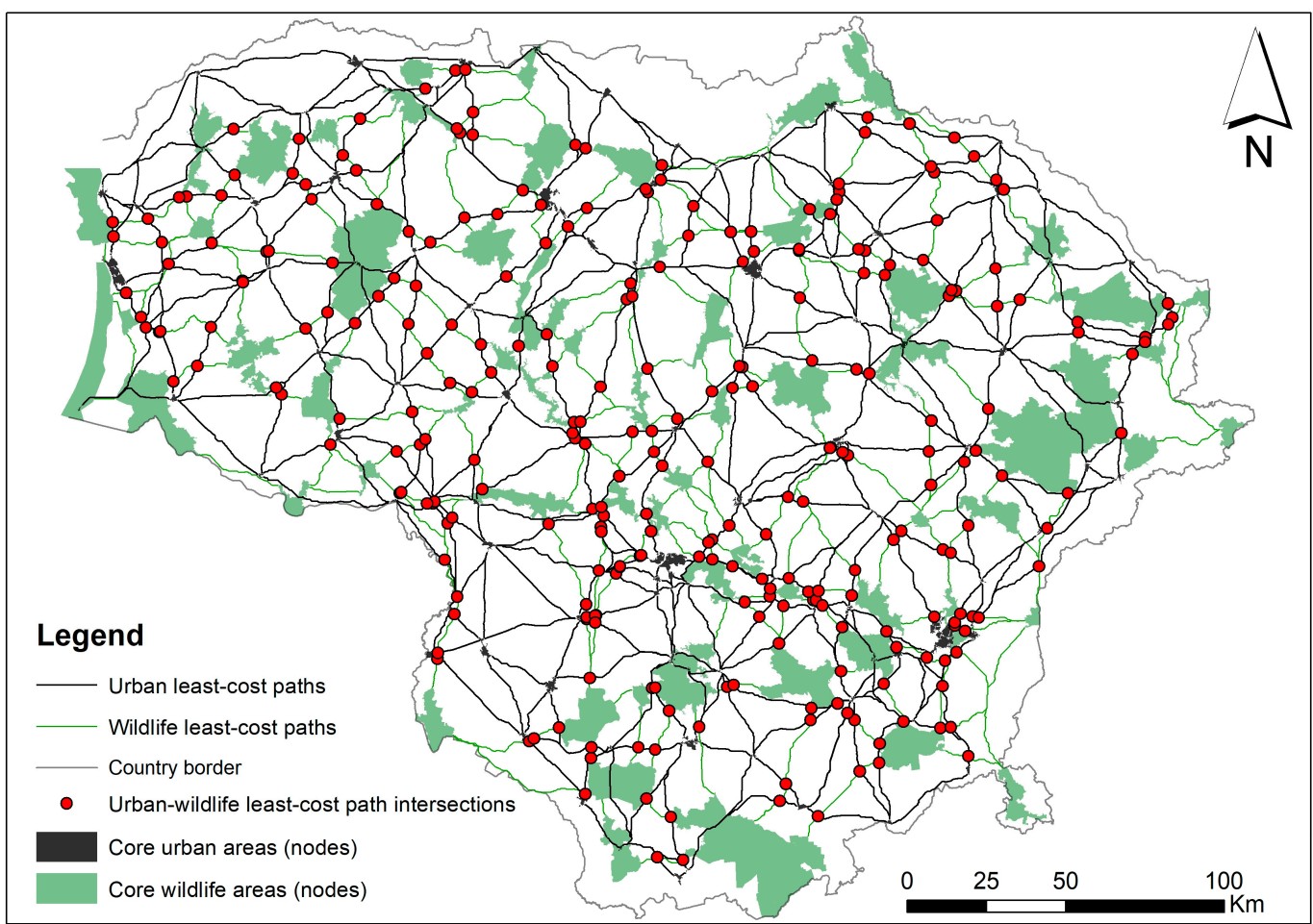

**Figure 5.** Intersections of urban and wildlife least-cost paths (corridors).

### 3.3. Ranking of Road Sections

The state-owned main road network (Figure 1) was divided into 77 sections (45 full-length sections of 25 km length and 32 shorter sections that range from 3.4 to 24.9 km length) starting from each road starting point. Each road section was characterised by the number of urban–wildlife least-cost path (corridor) intersections (within a 25 km buffer) using a double-counting approach (Figure 6). Correlation analyses show a strong positive relationship (r = 0.75) between the numbers of urban–wildlife least-cost path intersections and the numbers of mammal–vehicle collisions across main road sections. This suggests

that urban–wildlife least-cost path (corridor) intersection analyses are feasible and can support collision mitigation implementation strategies.

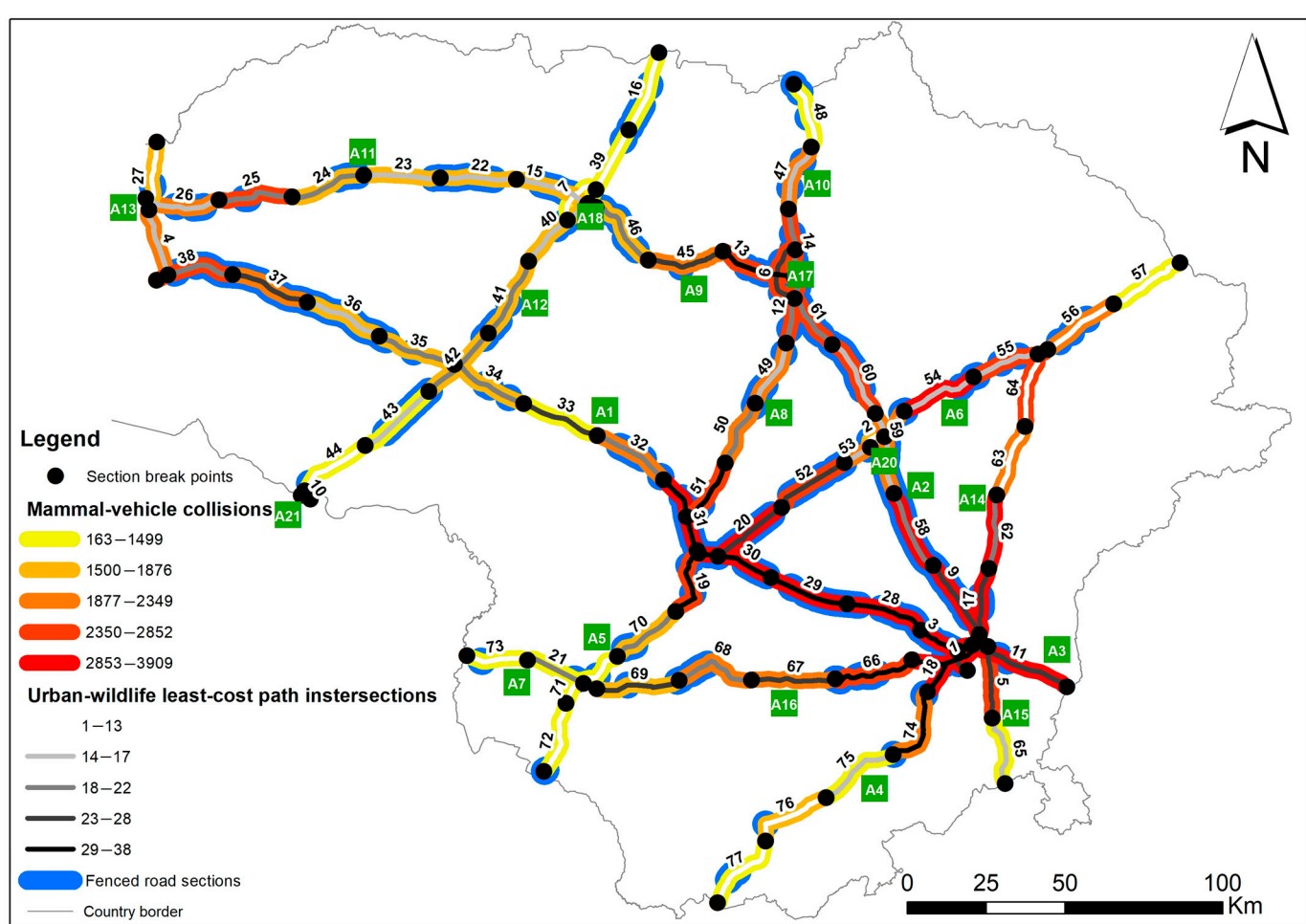

**Figure 6.** Urban–wildlife least-cost path intersection and mammal–vehicle collision quantile map. Road section identification (id) labels are placed above each road section, while main road numbers are placed within the green-coloured rectangle.

The road sections with the highest numbers of mammal–vehicle collisions and urban–wildlife corridor intersections are located in the western and southeast/central parts of the country and near main cities. Seven road sections within the highest urban–wildlife corridor intersection and mammal–vehicle collision quantiles are found on: road A1 (section ids: 3, 28, 29, 30, 31), road A16 (section id: 1), and road A4 (section id: 18). Road sections that fall within the lowest urban–wildlife corridor intersection and mammal–vehicle collision quantiles were found within the terrestrial border regions of the country (Figure 6).

The TOPSIS ranking scores (ranging from 0.0 to 0.89) were calculated for each road section (Figure 7) within the study area. Higher TOPSIS scores show better fit to the utility functions (maximisation) defined.

Road sections with the highest TOPSIS ranking scores are located in the western and central/southeast parts of the country, in the vicinity of main cities. Only two of the road sections (id: 58 and 59) that are on the road A2 received the highest TOPSIS scores. Thus, 15 road sections (Figure 7) are found within the highest TOPSIS quantile on: road A1 (section ids: 28, 29, 30, 31, 36, 37, 38), road A11 (section id: 22), road A12 (section id: 43), road A16 (section id: 66), road A2 (section ids: 9, 58, 59, 60), and road A6 (section id: 20).

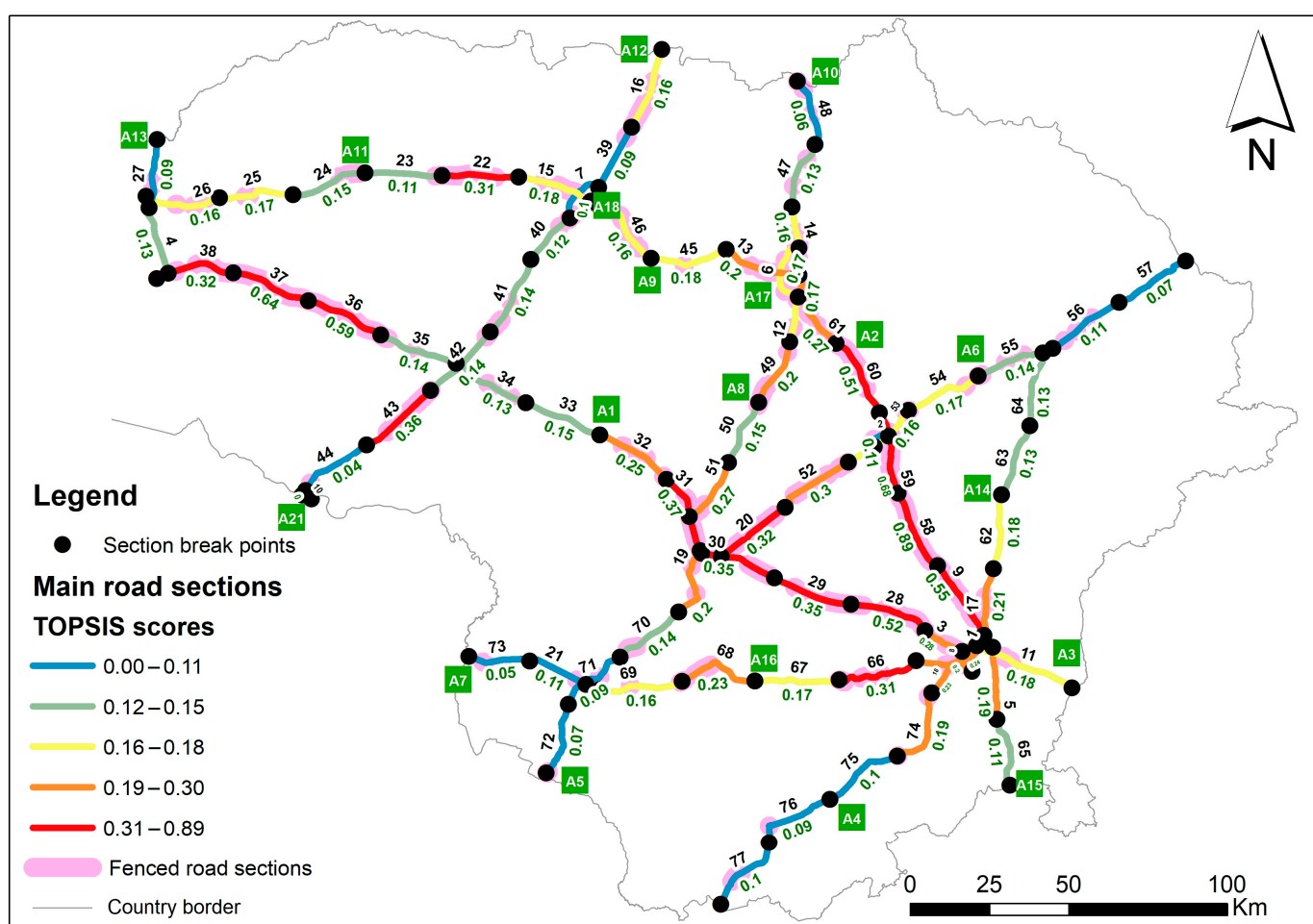

**Figure 7.** The road section ranking score quantile map. Road section labels (black labels) are placed above each section, while TOPSIS ranking scores are placed below road sections (green labels). Road numbers are on rectangular green labels shown as references.

Road sections that fall within the highest TOPSIS score quantile can be considered for mitigation with the highest priority. Installation of the missing parts of fencing and the addition of permanent crossing structures (such as overpasses) on these road sections would be the most beneficial for both urban and wildlife interfaces as well as being the most cost-effective because of the present fencing infrastructure there (Figure 7).

## 4. Discussion

### 4.1. Corridor Identification, Characterisation, and Ranking Framework

For the identification of wildlife corridors, we chose to use a circuit-theory-based least-cost path (corridor) identification paradigm [51] that uses land-cover-/land-use-based wildlife resistance grids. The circuit theory was originally designed for the identification of wildlife corridors [51]. In our study, we used it for the identification of urban corridors as well. Thus, land-cover/land-use resistance and conductance in both (urban and wildlife resistance grids preparation) cases are characterised by land-cover/land-use suitability and permeability scoring values.

To assure the randomness of the analyses, we split the main road network (characterised by the highest speed and traffic intensity) into road sections. Then, we characterised it with the number of mammal–vehicle collisions, the total length of wildlife fences, and the number of urban–wildlife least-cost path (corridor) intersections. Finally, we used multi-criteria analyses that allowed for us to identify road sections with the largest wildlife fencing, highest number of mammal–vehicle collisions, and highest number of urban–

wildlife corridor intersections (simultaneously). The road sections with the highest-ranking scores are the sections where permeant fencing and crossing structures shall be installed first. This road section identification, characterisation and ranking approach that we propose allows for the tight-linking of urban–wildlife corridor intersections and mammal–vehicle collisions and can reinforce locations for mitigation measures.

### 4.2. The Interface between Urban Sprawl and Wildlife

Wildlife–vehicle collisions are costly, often lethal for the animals, and can be lethal for vehicle drivers as well. Knowing where and how to best deploy mitigation measures on main roads with high traffic volumes is therefore valuable. There are different methods to identify wildlife–vehicle collision hotspots, which include, but are not limited to, roadkill data clustering [79–82], road shape [83], and road verge [84,85] analyses. However, the listed methods mainly focus on punctual and local data clustering approaches [24,86–88].

Additionally, wildlife–vehicle collision clusters are not stable in time and space [23]. However, the results of this study show that those urban–wildlife corridor intersections that are more stable in time and space can help relate less stable and locally identified collisions and reinforce the mitigation locations. The urban–wildlife least-cost path (corridor) intersections do not necessarily mean that an accident will occur in a particular location, but they do show a higher and more permanent risk for the accident to occur. The mammal–vehicle collision data used in this study confirmed that urban–wildlife corridor intersections have a strong relationship with mammal–vehicle collisions.

The ranking of road sections allows for the identification of the main mammal–vehicle-collision-prone road sections in Lithuania. The top-ranked road sections show simultaneously the highest number of mammal–vehicle collisions, the highest number of urban–wildlife corridor intersections, and the highest presence of fenced road sections (Figure 7).

It is important to note that the top shortlisted set of 15 road sections (Figure 7) identified in Lithuania should be mitigated simultaneously; otherwise, the ecological effect of mitigation might be negligible [89]. By law, in Lithuania, crossing structures and fencing are recognised as part of the road (such as bus stops, road signs, safety infrastructure, etc.) and can be installed only during certain road reconstruction projects. Afterwards, lower-ranked road sections can be mitigated at a later stage, using a step-wise corridor identification, characterisation, and ranking approach fuelled with new mammal–vehicle collision data.

Our results suggest that the largest share of urban–wildlife corridor intersections and mammal–vehicle collisions occur on those main road sections where fencing is already present (Figure 7). Fencing is a very effective collision mitigation measure, but only if it is properly installed and maintained [25,41,53,90–92]. Installing multispecies (mammal species) overpasses and closing the fencing gaps on highly ranked risky road sections may help to lower the rate of accidents without interfering with wildlife pathways.

Urban and linear infrastructures are the main drivers of wildlife–vehicle collisions. Roads and other linear infrastructure fragment habitats disconnect populations and create barriers to animal movement, thereby reducing animal dispersal and gene flow and directly increasing mortality [11,87,93]. Wildland intertwined with the urban interface creates tensions; from this, wildlife–vehicle collisions can be considered part of the ecological and habitat fragmentation caused by linear infrastructures [11,94,95]. This was the main reason why urban and wildlife corridors were analysed in a tightly integrated interface using a land-cover/land-use suitability and permeability classification strategy that was newly developed in this study. Land-cover suitability and permeability scoring-based resistance grids were used to identify corridor-land-cover connectivity, and this way, to identify wildlife and urban corridors and anchor collision locations to urban–wildlife corridor intersections. Our study confirms the results of another study [96], which found that landscape connectivity should be taken into account when planning mitigation measures.

We believe that the urban and wildlife corridor fusion approach used in this study allows for the identification, characterisation and ranking of road sections, and raises the awareness of the sustainable coexistence of human infrastructure and biodiversity.

As such, the results make a useful contribution to the potential reduction in mammal–vehicle collisions and can serve as guidance for the sustainable development of Lithuania's road infrastructure.

### *4.3. Limitations of the Study*

The mammal–vehicle collision data used in the study (Figure 1) may, however, not account for all accidents, as reporting of small accidents with wildlife to the authorities is not mandatory in Lithuania. However, reporting of wildlife–vehicle collisions when animals or people involved in the accident were killed or injured or where vehicles or road infrastructure were damaged is mandatory.

The study shows a generic wildlife and urban corridor identification approach based on the preparation of resistance grids and least-cost path intersection analyses. The variation among habitat requirements and movement abilities of different mammal species across the land-cover classes was not implemented in this study. However, the corridor identification approach is flexible and allows for the creation of resistance grids using different land-cover suitability and/or permeability scoring techniques that can be focused on species-specific movement patterns as well as urban development directions.

The ranking of road sections was currently limited to three criteria only. However, the list of criteria (including, but not limited to, species-specific criteria) can be extended, while criteria weights can be defined using different expert-based inputs.

This study was limited to the analysis of the road network. The identification of railway sections for the installation of permanent wildlife crossing structures remains an important part of ad hoc analyses because we had no mammal–train collision data. Nevertheless, the proposed main road sections for the installation of permanent wildlife crossing structures might strongly contribute to the preparation of a long-term accident-mitigation-focused strategic road and wildlife network design in Lithuania, which is so far lacking. The type and size of multispecies fencing and crossing structures depends on different animal species and their habitats in the vicinity of the roads [21,41,69,84,85]. However, the identification of crossing types and the precise locations of it on the main road sections and the evaluation of its efficiency remain future research challenges.

## 5. Conclusions

1.  Wildlife–vehicle collision locations have no spatial nor temporal stability due to changes in wildlife populations and urban sprawl, including, but not limited to, collision mitigation measures. The proposed urban and wildlife corridor identification method allows for anchoring collision locations at wildlife–urban corridor intersections and supports the planning of collision mitigation measures.
2.  Smaller spatial-scale land-cover data refined with medium-large spatial-scale 3rd party data and classified by land-cover suitability and permeability scoring values allow for us to identify urban and wildlife corridors and their intersections and indicate locations of potentially stable urban and wildlife conflicts.
3.  Correlation analysis showed a strong relationship ($r = 0.75$) between the numbers of urban–wildlife corridor intersections and mammal–vehicle collisions (2002–2020) within main road sections, meaning that the proposed corridor identification method is feasible and can support accident mitigation implementation strategies.
4.  Multiple criteria decision-support techniques allowed for us to identify 15 main road sections in the simultaneous presence of (i) the largest length of wildlife fences, (ii) the highest numbers of urban–wildlife corridor intersections, and (iii) the highest numbers of mammal–vehicle collisions. Highly ranked main road sections, if properly maintained, can support the sustainable coexistence of urban and wildlife interfaces as well as be the most cost-effective due to the presence of fencing infrastructure there.

**Author Contributions:** Conceptualization, A.K. and L.B.; methodology, A.K.; software, A.K.; validation, A.K.; formal analysis, A.K.; investigation, A.K. and L.B.; resources, L.B.; data curation, L.B.; writing—original draft preparation, A.K.; writing—review and editing, L.B.; visualization, A.K.; supervision, C.L.; project administration, C.L.; funding acquisition, C.L. All authors have read and agreed to the published version of the manuscript.

**Funding:** This research received no external funding.

**Data Availability Statement:** Not applicable.

**Acknowledgments:** We sincerely thank Edwin Schaaf and Benedikt Herrmann for their comments and linguistic revision, Boyan Kavalov for administrative support, and anonymous reviewers for their comments and suggestions that have resulted in a much-improved version of this manuscript.

**Conflicts of Interest:** The authors declare no conflict of interest. The views expressed are purely those of the authors and may not in any circumstances be regarded as stating an official position of the European Commission.

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
