# Peer review of "Identification of Urban and Wildlife Terrestrial Corridor Intersections for Planning of Wildlife-Vehicle Collision Mitigation Measures"

_land, doi:10.3390/land12040758_

Round 1

Reviewer 1 Report (Previous Reviewer 2)

See attached file

Author Response

Reviewer 2 Report (Previous Reviewer 3)

Dear authors,

thank you for the thorough revision of the manuscript; it reads much clearer now and the conclusions are derived from the study. I only have a few minor editorial comments left (line numbers referring to the Track-changes version):

line 27-29: suggest to rephrase: “... the road sections where wildlife fencing is, simultaneously, the longest, and the number of mammal-vehicle collisions and the number of urban-wildlife corridor intersections are highest. The results show...”

Table 1:  I suggest to remove all small mammals to a Footnote (up to the size of squirrel) without numbers because these numbers cannot be seriously taken; they are just occasional reportings. Please restrict the numerical data on medium-sized and large mammals only.

Lines 244-245: “... cell were summed up...  grid was created ...”

line 274: “larger than average size urban” – add “areas”? (something is missing here); also I just check that we are talking about arithmetic means (‘average’) in this paragraph, not medians (?)

Line 332: them -> these

Line 713: delete “and”? (the sentence does not read)

Author Response

This manuscript is a resubmission of an earlier submission. The following is a list of the peer review reports and author responses from that submission.

Round 1

Reviewer 1 Report

Improving the way we identify and prioritize sites and solutions for road impact mitigation is important, and undertaking such endeavor at the national level is impressive. The authors have clearly put a lot of work into this. I do think the manuscript could be a valuable contribution provided major changes are made to the language and organization of the paper, as well as to the rationale (why this particular method in the context of existing methods), and a much clearer explanation of the methods. Below are some more specific comments that I hope will help the authors polish this manuscript. 

Title

It reflects the content of the paper well, but the ‘urban corridor’ concept remains unclear throughout the paper. Depending on the response of the authors to those comments this may or may not have to change in the title. 

KEYWORDS

  • I would drop ‘main roads’ 
  • Urban and wildlife corridors are already in the title, so those can be dropped too
  • “Road impact mitigation” or something along those lines would be good to add. 

Abstract

  • 10 Roadkills and other issues of road ecology - change to ‘roadkills and other impacts of roads on wildlife’ 
  • 12: causes (instead of reasons)
  • 11-14 - confusing - talks about the variability of roadkill hotspots versus permanent sites for fencing and crossing structures. 
  • 17 -  A strong relationship
  • 18 - avoid starting a sentence with a number
  • 21-23 - See Kintsch et al. https://arc-solutions.org/wp-content/uploads/2021/03/08.-ICOET-WildlifeCrossingGuilds-paper.pdf, and https://www.sciencedirect.com/science/article/pii/S0195925522002761 
  •  

Introduction

General:

  • I would reorganize the introduction a little. All the information is there, but it needs better flow. 
    • The first paragraph: focus on the expansion of road networks and how this changes the interface between natural habitat and the urban environment. 
    • The second paragraph can go into more detail on wildlife vehicle collisions as one of the effects of those expanding networks or road upgrades. 
    • The third paragraph can address measures taken to address WVC (currently mostly in the second paragraph), which then leads nicely into the need to identify where to install these measures or which measures to choose. 
    • The fourth paragraph can talk a little about the methods used to identify sites for mitigation measures.
    • The last paragraph are your questions. 

Considering identification of roadkill mitigation sites has been done with various methods, I recommend including this in the introduction with an explanation why a different method is being applied here. Circuit theory has been used for this, but only for wildlife movements. It is not clear how the ‘urban corridor’ aspect adds to the identification process. 

Specific

  • 28-29: omit ‘in numerous countries’. It’s a global concern
  • 32: as the wildland-urban interface expands
  • 32: ‘current roads’ may make the sentence more easy to read
  • 33: are considered part of (omit ‘as’)
  • 35: the sentence is a bit redundant, and ‘by mitigating WVC’ can be omitted.
  • 37: not clear what ‘passages’ mean here. ‘Bridges, culverts and tunnels would fall into the category of over-and underpasses. ‘Poles’ would have to be explained further. Maybe it would be good to group these strategies and provide a little bit more detail on each. 
  • 39: types of measures
  • 44: why separate ‘underpasses with water flow’ from viaducts and large mammal underpasses? Both can have water flow. I think the categorization needs some work. 
  • 48-49: this loops back to the first paragraph. This would make a good introductory sentence to that first paragraph. 
  • 57-62 - this needs to be explained better. Fencing also comes with serious implications, which need to be addressed here. E.g. changing to migration routes of wildlife can lead to increased calorie expense with lower reproduction. (Eg. https://movementecologyjournal.biomedcentral.com/articles/10.1186/s40462-022-00336-3
  • 61 - “higher category roads” - not clear. Best to leave this out and just refer to roads with higher traffic volume and speed 
  • 67-68 - This needs further explanation. If ‘urban corridors’ are really just infrastructure as defined in line 64, why do they need further identification? It appears that a very longwinded process was used to identify them (from the methods). In the introduction, it must be explained why - what are these urban corridors, what aspects need to be identified, how is this done (generally speaking, not a full method)? I think if this bit of the introduction can be elaborated upon, the rest of the paper will make a lot more sense. 
  • 72 - The aim of this research was 
  • 73 - in the vicinity
  • 74 - not clear what FCS stands for

Materials and methods

General

From the methods, it is difficult to understand what you intended to do. It may be good to have a paragraph where you state that you identified wildlife corridors, that you looked at the intersection of wildlife corridors with infrastructure and that this was then compared with roadkill data (which is what I am currently thinking you did, but it is not sufficiently clear whether this is indeed the case). Next, you can give an overview of the spatial layers used and how you classified them. It is important to explain why you used such classification. Based on table A2, it seems like there are overlapping categories and many that would not be necessary. What are your choices based on? Habitat use of the species included? Any guidance by data from the literature? In table A2, it seems like permeability for wildlife was scored generally and not on a per-species basis, which we know is incorrect. If you want to use a multi-species approach, it is still important to consider the individual species (group) characteristics. 

I understand why you may want to use acronyms, but there are so many in this paper that the reader gets lost and has to repeatedly go back. If you want to keep the acronyms, it may be good to include a list of them, so the reader has a central place to go back to. If the idea is that other projects also use this method, a diagram of the steps would be helpful. 

Specific

84 - what do you mean by ‘of national significance’? Did you exclude some roads that only connect small towns, for example? This must be clarified. 

99- from both ends of the fence

99-100 - You say you used more information. Where did that information come from? What information was this?

100-102 - Move to discussion in a paragraph on limitations of the study (same for the repeated 111-115, which seems to contradict 100-102)

107 - WVCs?

95-99 - This is repeated with different years and numbers in 109-111

116-120: it may be helpful to include a table of the layers, sources and scales

147: It is not clear why identification of urban corridors requires anything more than a map that shows the roads. As described in line 64. Maybe my interpretation of ‘urban corridor’ is incorrect. In either case, it would need to be clearly defined, and it will need further explanation as to why a model was used to identify them. Overall, it is not really clear why you would do anything more than model wildlife movements across roads, since this paper is focused on roadkill. 

191-193: Software requires references

RESULTS

General

With the methods not being very clear, it is not easy to interpret the results. I made a few specific comments in this section, but I think it will greatly improve with updated methods, and moving the paragraphs from the results that are really methods into their appropriate section. 

Specific

195-212 - This appears to belong in the methods, rather than the results, as it describes what you did, not what you found. Figure 2 is the result of what is described here. 

227-229 - Methods (or it could be included in the discussion). Regardless, it needs further explanation. Are all ‘core wildlife areas’ protected areas? If so, I did not see that explicitly stated anywhere. What type of regulation for roads through these areas exist? Have they been effective? If this is not the fencing you are proposing here, I would argue that it is worth including them and in the discussion explain what has been done and what is or is not effective in terms of regulations currently in place. 

231-233 - Move to methods and explain the correlation analysis. What program did you use? What was correlated? Note that roadkill hotspots are generally studied with Ripley's K statistic (e.g. in software like Siriema). See for example: https://revistas.ufrj.br/index.php/oa/article/view/8254 . A correlation coefficient needs to be reported with its degrees of freedom and p-value. 

235-237 - Move to discussion. 

243 - Sentence should not start with a number

246-249 - It would be helpful to explain the TOPSIS method in more detail in the methods, so that the reader can easily interpret the results and their meaning. ‘Machine based criteria weights’ is very vague. It is difficult to understand what this result means. 

305-312 - From this, I would understand that existing fencing has not been an effective strategy against wildlife-vehicle collisions and that you are recommending that wildlife underpasses are used as a supplemental strategy. 

313 - Spacing of existing fencing? Please clarify 

313-321 - It is not clear what message you are trying to convey here. What projects indicate the distance between wildlife crossings? Why are those distances not the same as the distance between wildlife bridges? Are there other crossing structures involved? Or is the first part referring to key sites where wildlife may cross, and the second part to structures that permit save crossing? Please clarify. 

Figure 2

The legend does not clearly show what the values represent. What is the ‘index’? Does it represent resistance? If so, why only from 1-22, as resistance is generally expressed on a range from 0-100?

Figure 3

It is not clear what this represents, but this is due to the unclear explanations in the text. The color schemes are confusing, because red represents very high in one and very low in the other. It is not clear what ‘pinch points’ are. The core areas

Figure 4

Borderline —> Country border 

Figure 5

It is not clear what these ‘quantiles’ are referring to. Are they numbers of wildlife-vehicle collisions? What are the cut-off points for these quantiles? Why are those the cut-off points? 

Figure 6

From the methods, it is not clear what these TOPSIS scores mean. Why do they range from 0-0.89? What do they represent? Why are they cut off at uneven intervals? 

DISCUSSION

General

The discussion must focus on the interpretation of the results and not justify the need for the research or include generic solutions from the literature, as those belong in the introduction. 

Specific

265-286 - Introduction. The first paragraph in the discussion should summarize the objectives and major findings. 

Table A2

The classifications seem odd, and there is no explanation as to why these classifications were chosen. 

  • What is ‘urban fabric’?
  • What are ‘associated lands’ for the roads?
  • Dump sites: landfill site?
  • Is irrigated land not overlapping with some of the other categories, like any of the forms of agriculture mentioned below? 
  • Sport leisure facilities - why is this relevant? 
  • What would classify as ‘sparsely vegetated area’? Is this a natural habitat, human-altered habitat? 

Why not include 

Figure A1

This is a really helpful figure to get a general overview of the situation. 

The difference between major city boundaries and all build-up areas is not clear. 

What is the difference between a core wildlife area and a protected area? What determined the overlap? Please explain in more detail in the methods. 

Table 1 and Table A1

I would keep these together so that the reader can see all species at once. These can all go into an Appendix, with brief mention in the text of top three species, for example. 

Some additional references to consider

  • If looking at roadkill and connectivity, these must be included: 
    • https://link.springer.com/article/10.1007/s10344-018-1241-7
    • https://link.springer.com/article/10.1007/s10531-016-1194-7
    • https://link.springer.com/article/10.1007/s10344-010-0478-6 

  • This paper must be considered in the concept and conclusions of the manuscript: 
    • https://besjournals.onlinelibrary.wiley.com/doi/full/10.1111/1365-2664.12870

Reviewer 2 Report

See attached file

Reviewer 3 Report

This paper is interesting because it tries to distinguish the road sections with the most effective mitigation potential of wildlife-vehicle collisions at a country scale. I generally find the approach interesting and the analysis and results worth intruction to a wider research community. However, I had many problems with the presentation, which are more or less marked on the attached manuscript (not all language problems though – too much work, sorry!). Below I just summarize my main points; please check the attached file for details.

#1  The main problem is that the analytical techniques have not been described and, thus, the whole result is actually not transparent and the analysis in not repeatable. I hope this can be easily corrected by just adding concise and accurate descriptions in Ch. 2.4 and 2.5, but please be really careful with this. Specifically, how did you calculate (i) “estimates of the net movement probability of a random wanderer through a given cell” (line 153); (ii) “wildlife and urban pinch points (bottlenecks)” (line 166), (iii) „machine based weighting backed technique for order of preference by similarity to ideal solution (TOPSIS) method“ and what does “utility function set to maximisation” mean. You must place yourself to a position of an expert reader who should be able to repeat this!

#2 The text is extremely difficult to read because of hundreds of abbreviations, even in Conclusions section. My suggestion is to retain only those that are frequently used or right in the same place (e.g., paragraph). Particularly in Discussion, make a difference between the concept and the variable, i.e when you talk about denser traffic, it is a phenomenon that is not restricted to your “AADT” (I made examplary corrections in the 1st paragraph of Ch. 4.2).

#3 Most of the Discussion is not currently devoted to the results (4.1 entirely, 4.2 largely). If it stays so, it should be condensed at least two times. Please note that this is a research paper, not a expert opinion to managers. Furthermore, because as a reviewer I cannot check the background of most of the speculations provided (since they are not related to your analysis), I feel rather uncomfortable for such text passing the peer review. So, please leave out all passages that are not related to your data, retain only fact-derived lines of thoughts, and condense the text to be to-the-point.

#4  At lines 128-138 you present your scoring system for wildlife (and urban) habitats, but I kept wondering whether there is such one thing as "wildlife habitat", with a specific permeability etc. So please be more explicit: I suspect you mean mammals, and perhaps even some model species here. It is nothing wrong if you did so, but a limitation is that there is much variation among habitat requirements and movement abilities of different species across the landscape. This should be addressed as a study limitation, meaning that some other model system may have yielded a different prioritization of the road sections.

#5 Another possible limitation I encountered at line 144 is that I understand that every 20*20 m cell in your grid got a value independently of the surrounding cells (?). This may be unrealistic since both humans and wildlife may avoid hostile surroundings or require minimum patch size.

#6 Since your raise the point in Introduction that your approach might be useful also in other countries, please elaborate this in Discussion. What exactly might be learned?

#7 Line 191 says that you prepared “quartile maps”, but what does this mean? I found “quantile” under figures, and the legends do not match with quartile division...

#8 Lines 227-229 seem to suggest that you left out large wildlife core areas because you equalized these with protected areas (?). I cannot understand this argument or why you did not use the borders of actual protected areas to take this into account.

#9 I could not follow the conclusions on using smaller spatial scale for such analysis. To my understanding you never took an alternative approach to compare; thus you cannot logically conclude that a smaller scale is better.

#10 Please recheck the language throughout. I tried to help with many corrections but some more is needed.
